# Urban modeling of shrinking cities through Bayesian network analysis using economic, social, and educational indicators: Case of Japanese cities

**Haruka Kato** * *

Department of Housing and Environmental Design, Graduate School of Human Life and Ecology, Osaka Metropolitan University, Osaka, Japan

* haruka-kato@omu.ac.jp

## Abstract

Shrinking cities due to low birthrates and aging populations represent a significant urban **planning** issue. The research question of this study is: which economic, social, and educational factors affect population decline in Japanese shrinking cities? By modeling shrinking cities using the case of Japanese cities, this study aims to clarify the indicators that affect the population change **rate**. The study employed Bayesian network analysis, a machine learning technique, using a dataset of economic, social, and educational indicators. In conclusion, this study demonstrates that social and educational indicators affect the population decline rate. Surprisingly, the impact of educational indicators is more substantial than that of economic indicators such as the financial strength index. Considering the limitations in fiscal expenditures, increasing investment in education might help solve the problem of shrinking cities because of low birthrates and aging populations. The results provide essential insights and can function as a planning support system.

## 1. Introduction

### 1.1 Background

Shrinking cities pose significant urban planning issues [1]. These issues have been seen, for example, in former East Germany due to political changes [2] and in United States' Rust Belt due to the economic decline [3]. In addition, during the COVID-19 pandemic, it was reported that city populations declined due to urban exodus [4]. Among the various types of population decline, this study focused on Japanese shrinking cities due to the low birthrates and aging populations [5]. The population decline is inevitable in Japan, where immigration is low. In 2021, the Japanese population aged over 65 comprised 36.21 million people, constituting 28.9% of the total population [6]. It was also reported that the national population will decrease to between 38 and 65 million people by 2100 [6]. Fig 1 shows the population change rate (*PCR*) in Japanese cities in 2010. In Fig 1, PCR is the percentage of change from population in

Education, Culture, Sports, Science, and Technology (MEXT) Institutional Data Access (contact via MEXT, +81-5253-4111, https://www.mext.go.jp/b_menu/toukei/001/1322611.htm) for researchers who meet the criteria for access to confidential data.

**Funding:** H.K., 21K14318, JSPS KAKENHI, https://kaken.nii.ac.jp/en/grant/KAKENHIPROJECT-21K14318/ The funders had no role in study design, data collection and analysis, decision to publish, or preparation of the manuscript.

**Competing interests:** The authors have declared that no competing interests exist.

**Fig 1. Population change rate in Japanese cities in 2010.** In Fig 1, left side map indicates the location of Tokyo and Osaka in major East Asian cities, and right side indicates map focused on the Osaka Metropolitan area. Fig 1 shows the population change in graduated colors from red to blue. The gray color areas are other cities, such as merged cities. Republished from Fig 1 under a CC BY license, with permission from ESRI Japan, original copyright 2023.

2005 ($P_{2005}$) to population in 2010 ($P_{2010}$), as calculated in Eq (1). Fig 1 indicates that most cities' populations have been declining. Similar problems are likely to be faced in other East Asian countries such as China [7, 8]. In China, it was identified that 153 cities were shrinking between 1992 and 2019 [9]. Population decline driven by low birthrates and aging populations poses various problems, including overall economic and social decline [10]. For example, population decline correlates with various types of social problems, such as economic decline and increasing vacant houses [11]. However, there is a cascading and complex relationship between economic and social factors within the context of population decline, and researchers have yet to systematically elucidate the factors affecting population decline.

$$PCR = \frac{P_{2010} - P_{2005}}{P_{2005}} \times 100 \qquad (1)$$

The research question of this study is: which economic, social, and educational factors affect population decline in Japanese shrinking cities? These economic, social, and educational factors might be strongly or weakly interrelated with the PCR; such relationships are complex, ranging from the micro to the macro level [12]. That means that the relationship would not reveal deductively using a fixed framework, but inductively using a large set of indicators. The relationships between these factors also do not occur through a simple linear mechanism, but through a non-linear and complex mechanism [13]. To understand these relationships, the logical framework to be utilized is an inductive statistical urban model using various indicators of shrinking cities [14]. The statistical urban models allow policymakers to monitor population decline [15]. For the analysis, this study developed an urban network model that interrelates economic, social, and educational indicators. By providing systematic insights into the issues emerging due to population decline, the model can deliver a planning support system for municipal leaders and urban planners.

## 1.2 Purpose

This study aims to clarify the indicators that affect the PCR by modeling shrinking cities. For the analysis, this study uses the case of Japanese cities. In Japan, there were 1,733 cities in 2010. Among these, this study analyzes 1,316 cities whose populations were declining as of 2010 (Fig

1). Japan is suitable for this study because it contains the largest number of shrinking cities due to low birthrates and aging populations worldwide. In Japan, the government has begun to develop urban plannings for shrinking cities [16]. Therefore, this study's results from Japan may contribute by providing insights and a planning support system for shrinking cities worldwide facing similar problems.

Methodologically, this study employed Bayesian network analysis, a machine learning technique, to model shrinking cities using a dataset of economic, social, and educational indicators by cities. The Japanese Ministry of Education, Culture, Sports, Science, and Technology (MEXT) developed the dataset used in this study to promote research that contributes to policy formulation. The dataset comprises cross-sectional data of 259 indicators, including economic, social, and educational indicators [17]. The dataset covers most city-level government statistics that are available publicly. The Bayesian network constructed from this dataset is a stochastic model representing the quantitative causal relationship between individual indicators with conditional probability [18]. The probabilistic estimation of the network makes it possible to predict uncertain scenarios.

### 1.3 Literature review

The novelty of this study is to develop an urban statistical model of shrinking cities by Bayesian network analysis. Urban models of shrinking cities have been proposed in previous studies. In an early study, the researchers conceptualized an urban model consisting of economic and population decline and policy changes [19]. As research on shrinking cities progressed, particularly in Europe, researchers also conceptualized an urban model in which diverse forms of decline at the global and regional levels caused population declines [20]. These conceptual urban models indicated that population decline is related to economic and social decline. However, it was also found that the indicators associated with population decline are multifaceted and pertain to diverse perspectives [21]. Therefore, to investigate the complicated relationships among these indicators, researchers need to develop both an urban statistical model that strength conceptual models of shrinking cities. Accordingly, this study aimed to develop an urban statistical model of shrinking cities.

Previous studies have analyzed the statistical relationship between population decline and economic indicators in shrinking cities. Specifically, the positive correlation between population change and industrial diversity was clarified [22]. For example, it was investigated the case of Yubari City, a single-industry mining city in Japan, well-known for suffering from rapid population decline and financial collapse because of the mining industry decline [23]. From a different perspective, researchers demonstrated the positive correlation between investment network centralities and population change in shrinking cities [24]. In addition, it was explained that development zone policies have helped curb population decline in these shrinking cities [25]. Academicians also demonstrated that shrinking cities are influenced by socioeconomic development and social indicators, including expenditure on education [26]. In Japan, it was found that population decline in densely inhabited districts is caused by the outflow of residents and the formulation of suburban residential areas [27]. In addition, in the case of Yokohama city, Japan, it was clarified that urban shrinkage was correlated with the aging population, distance to the nearest parks, and proportion of private houses and flats [28]. Despite the important contributions of these past studies, they failed to elucidate the complex and cascading relationships of factors related to population decline. Accordingly, the originality of this study lies in its extension of these prior pieces of evidence by developing an urban statistical model of shrinking cities using a dataset comprising economic, social, and educational indicators, and in its aim to identify the factors influencing PCR. For the analytical

methodology, this study refers to the study by Kato [29], wherein urban modeling developed by Bayesian network analysis was used to estimate future populations.

## 2. Materials and methods

### 2.1 Economic, social, and educational indicators

As mentioned above, this study's data come from a cross-sectional dataset of economic, social, and educational indicators by cities which MEXT collected to promote research contributing to the formulation of educational policies [17]. This study used this dataset because it covers most publicly available city-level government statistics in Japan. Specifically, it stems from sources such as the Japanese Census, the School Basic Survey, the National Survey of Academic Progress, the Social Education Survey, the Survey on Time Uses and Leisure Activities, the Housing and Land Survey of Japan, the Major Financial Indicators of Local Governments, the Economic Census, and the National Survey of Family Income and Expenditure [30]. While this dataset cannot be shared publicly because of governments' confidentiality, the dataset is available from the Japanese MEXT Institutional data access for researchers who meet the criteria for access to confidential data. Since April 5, 2022, the author has received permission from MEXT to provide this data. All indicators are listed in S1 Table.

The indicators present in the dataset are used not only for policymaking but also for academic research [31, 32], and the dataset encompasses 259 indicators in total [33]. This study used all 259 indicators containing numerical and nominal scale data. The dataset covers the ten years from April 1, 2001, to April 1, 2011. This study used data as of April 2011. Although the dataset is 10 years old, no other datasets in Japan have such a comprehensive range of indicators. As population decline is a long-term urban phenomenon, the dataset is valuable and appropriate for this study.

### 2.2 Bayesian network analysis

Bayesian network analysis was used for urban modeling based on the economic, social, and educational indicators. Compared to similar statistical analysis methods, such as structural equation model analysis, neural network analysis, and decision tree analysis, Bayesian network analysis allows for the flexible analysis of nonlinear and non-normal relationships between indicators [29]. It also enables researchers to obtain robust models that avoid collinearity risk [34]. It is important for the current study that that the analytical process of choice can avoid collinearity risk because 259 indicators are under scrutiny. This study used BayesiaLab 10.2 as its Bayesian network construction algorithm [34].

This study adopted the maximum weight spanning tree (MWST) and taboo algorithms for the optimal local search for each child node; the MWST algorithm was deployed first, followed by the taboo algorithm. The MWST algorithm makes it possible to compute big data, which this study analyzed, faster than other algorithms [35]. The taboo algorithm is effective for refining networks built by updating another network learned on a different dataset, because it refers to structural learning by implementing the taboo search for Bayesian networks [36].

For the analysis, this study set the indicator of PCR as the target variable; Bayesian network analysis revealed the total effect (TE) and correlation of indicators on the PCR. TE was analyzed by standard target mean analysis (STMA), which uses the mean value evidence to go through the indicators' variation domain and measure the impact of indicators on the target's mean. That is, TE is the derivative of the total effects curve computed at the a priori mean of that variable, $\delta_x = 0$ and $\delta_y = 0$ [34]. The standardized total effect (STE) normalized the TE by taking into account the ratio between the standard deviations of the indicators ($\sigma_x$) and the

target indicator (PCR) ($\sigma_y$) [34]. The STE was calculated using Eq (2), as follows:

$$STE = \frac{\delta_y}{\delta_x} \times \frac{\sigma_y}{\sigma_x} \qquad (2)$$

Correlation analysis was based on mutual information (MI), defined as the difference between the marginal entropy H(Y) of the target indicator (PCR) and its conditional entropy H(Y|X). The MI was calculated using Eq (3), which is equivalent to Eq (4); in the latter, $p(x,y)$ is the joint probability function of X and Y, while $p(x)$ and $p(y)$ are the marginal probability functions of X and Y.

$$MI(Y, X) = H(Y) - H(Y|X) \qquad (3)$$

$$MI(T, X) = \sum_{x \in X} \sum_{y \in Y} p(x, y) \log_2 \frac{p(x, y)}{p(x)p(y)} \qquad (4)$$

The binary mutual information (BMI) is the amount of information brought by each indicator to the knowledge of the state of the target indicator (PCR) compared to an unconnected network [34]. That means that BMI is amount of information brought by each variable to the knowledge of the state of the target indicator (PCR).

## 3. Results

Fig 2 shows the urban model of shrinking cities developed by Bayesian network analysis. In the network, Pearson's correlation coefficients between indicators are shown in the links. In total, 208 out of 259 indicators formed a significant network. These 208 indicators, including economic, educational, and social indicators, represent an interrelated urban model.

The lower part of Fig 2 shows a Bayesian network focusing on PCR as the target indicator. Among social indicators, "population change rate aged 0–14" and " population aged over 65" are strongly related to the PCR. "Population aged over 65" has a negative correlation coefficient, indicating that an increase in the number of older adults results in a decrease in the PCR. Meanwhile, relevant educational indicators include "number of children per teacher in elementary schools," "number of children per teacher in junior high schools," and "number of students per educational computer in junior high schools. "Financial strength index," which serves to represent the financial strength of a local government and is the average of the values obtained by dividing the standard financial revenue amount by the standard financial demand amount over the past three years, is one of the relevant economic indicators. Finally, urban planning indicators were not found to be related to PCR.

Table 1 shows the STE, MI, BMI, and p-value of each indicator in relation to the PCR. Table 1 focuses on 14 indicators with STE more than |0.2|. Hence, indicators with STE less than |0.2| have little effect on the PCR. The STE, MI, and p-value of all indicators are listed in S1 Table.

Table 1 shows that the most significant indicators of the PCR are the social indicators of "population change rate aged 0–14" (STE = 0.73), " population aged over 65" (STE = -0.70), "average age of unmarried men" (STE = -0.43), and "natural population change rate" (STE = 0.57). In shrinking cities with a PCR under -12.6, BMI is higher for "population change rate aged 0–14" (BMI = 0.35) and " population aged over 65" (BMI = 0.35), suggesting that these two indicators strongly correlate with increased population change. It is worth noting the low STE of the "population aged 15–65" (STE = 0.09), which was not included in Table 1. The strong relationship between the "population aged 15–65" and the PCR might suggest that

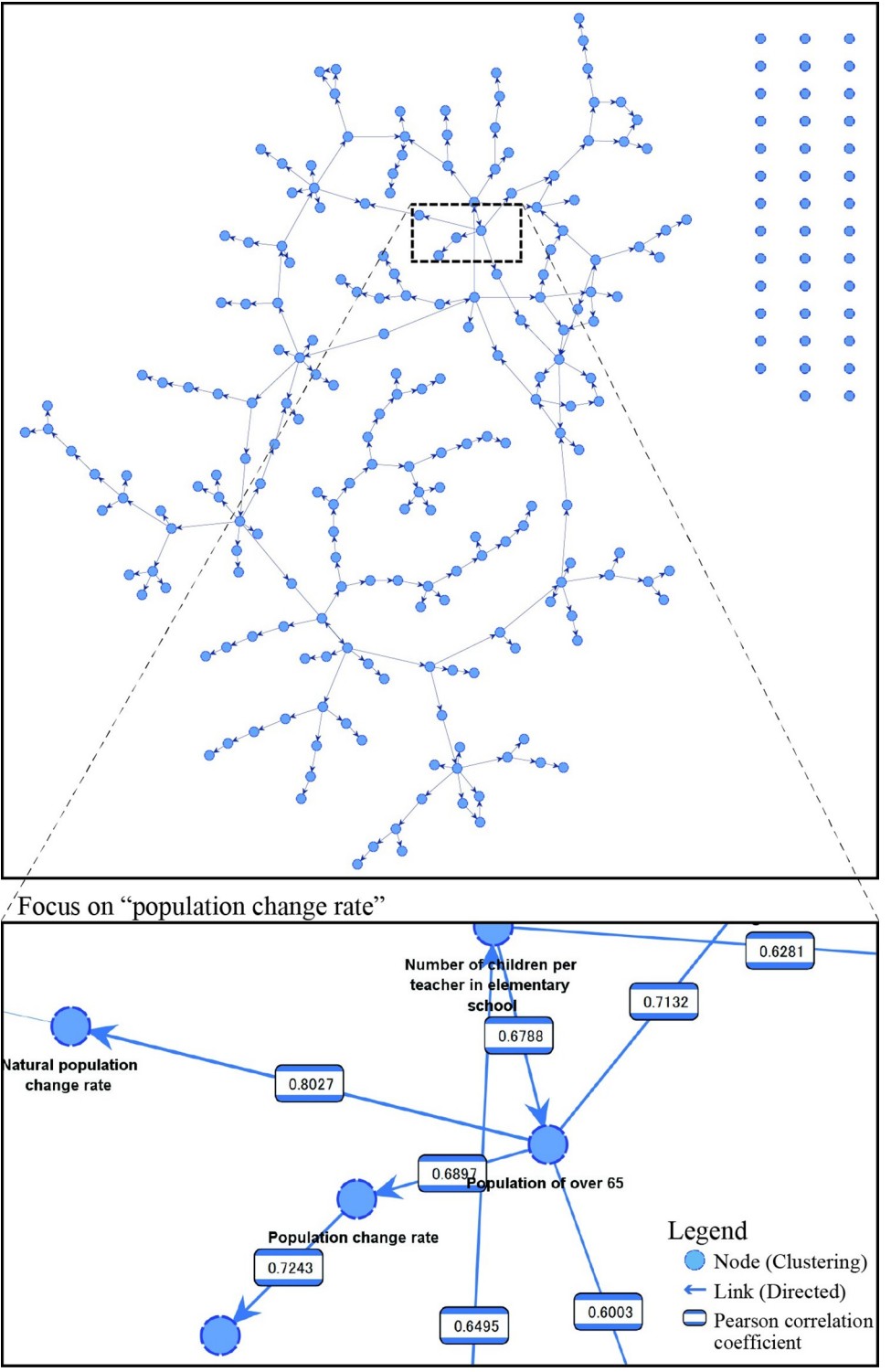

**Fig 2. Bayesian network of shrinking cities.** The upper graph shows the whole network. The lower graph shows the network around the PCR. In the lower graph, numbers between links indicate the Pearson's correlation coefficient.

**Table 1. STE, MI, BMI, and p-value of each indicator.**

| Indicators | STE | MI | BMI ≤0 | BMI ≤-4.0 | BMI ≤-7.7 | BMI ≤-12.6 | p-value |
|---|---|---|---|---|---|---|---|
| Population change rate aged 0–14 | 0.73 | 0.54 | 0.09 | 0.19 | 0.16 | 0.35 | ** |
| Population aged over 65 | -0.70 | 0.50 | 0.08 | 0.16 | 0.14 | 0.35 | ** |
| Natural population change rate | 0.57 | 0.30 | 0.05 | 0.10 | 0.05 | 0.22 | ** |
| Percentage of population aged 0–14 | 0.51 | 0.22 | 0.04 | 0.08 | 0.02 | 0.15 | ** |
| Number of children per teacher in elementary school | 0.47 | 0.20 | 0.03 | 0.07 | 0.03 | 0.15 | ** |
| Average age of unmarried men | -0.43 | 0.16 | 0.03 | 0.06 | 0.01 | 0.11 | ** |
| Number of children per teacher in junior high school | 0.36 | 0.10 | 0.02 | 0.04 | 0.01 | 0.08 | ** |
| Population change rate aged over 65 | 0.30 | 0.08 | 0.01 | 0.03 | 0.01 | 0.06 | ** |
| Financial strength index | 0.29 | 0.09 | 0.01 | 0.04 | 0.01 | 0.07 | ** |
| Percentage of combined classes in elementary school | -0.27 | 0.06 | 0.01 | 0.03 | 0.00 | 0.05 | ** |
| Average age of unmarried women | -0.26 | 0.05 | 0.01 | 0.02 | 0.00 | 0.04 | ** |
| Number of students per educational computer in junior high schools | 0.25 | 0.06 | 0.01 | 0.02 | 0.00 | 0.05 | ** |
| Number of students per educational computer in schools | 0.21 | 0.04 | 0.01 | 0.02 | 0.00 | 0.03 | ** |
| Percentage of workers in primary industry | -0.20 | 0.04 | 0.01 | 0.01 | 0.00 | 0.03 | ** |

Table 1 focuses on 14 indicators with STE more than |0.2|. In Table 1, STE is standardized total effects. MI is mutual information. BMI is binary mutual information.

\* indicates *p*-value < 0.05.

\*\* indicates *p*-value < 0.01.

the population has decreased due to migration, as in other shrinking cities in the US and European countries. However, the model indicates the weak relationship between the "population aged 15–65" and the PCR in Japanese shrinking cities. Thus, population decline in Japanese shrinking cities occurred primarily due to natural population change rather than population change because of migration.

The PCR is also found to be affected by educational indicators, with relevant factors including "number of children per teacher in elementary school" (STE = 0.47), "percentage of combined classes in elementary school" (STE = -0.27), and "number of students per educational computer in schools" (STE = 0.21). Therefore, population change is related to aspects of education, such as the number of teachers and educational computers.

Two economic indicators are shown to affect the PCR, namely "financial strength index" (STE = 0.29) and "percentage of workers in primary industry" (STE = -0.20), albeit other economic indicators such as "per capita income of prefectural citizens, and "taxable income" did not affect PCR. Hence, the analysis suggests that the PCR is more strongly influenced by social and educational indicators than economic indicators in Japan.

Many other indicators show a weak relationship with the PCR but are not listed in Table 1, including indicators related to urban facilities (e.g., "number of community centers" and "number of medical clinics,") and housing, (e.g., "number of homes located more than 500 meters from the nearest elementary school" and "number of land transactions").

Fig 3 shows the probabilistic change of shrinking cities with rapid population decline. In this study, shrinking cities with rapid population decline are those with a PCR under -12.6%, which accounted for 4.6% of shrinking cities in Japan. Among these cities, the probabilities of the social indicators of "population change rate aged 0–14 ≤ -26.3" and "natural population change rate ≤ -1.3" change from 4.1% to 52.7% and from 7.0% to 38.9%, respectively. Regarding educational indicators, the probability of "number of children per teacher in elementary school ≤ -7.8" and "number of students per educational computer in schools ≤ -4.0" changed from 17.8% to 55.1% and from 38.4% to 59.9%, respectively. Regarding economic indicators,

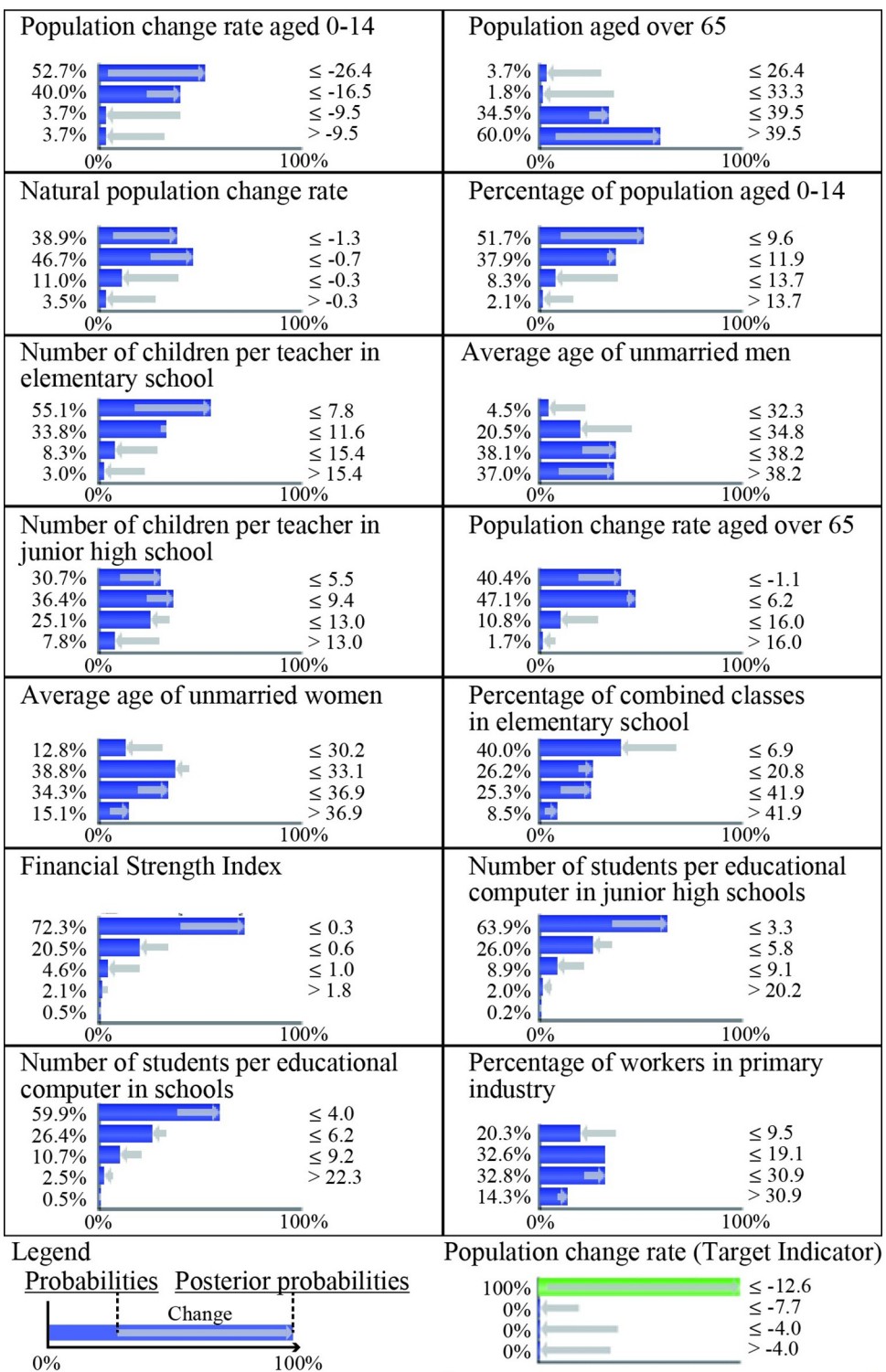

**Fig 3. Probabilistic change of shrinking cities with rapid population decline.** Fig 3 focuses on 14 indicators with STE more than |0.2|.

the "financial strength index $\leq 0.34$" probability changed from 40.7% to 72.3%. These results suggest that shrinking cities with rapidly declining populations experience a significant deterioration in social, educational, and economic indicators.

## 4. Discussions

In conclusion, this study demonstrates that social and educational indicators affect the population decline rate by modeling shrinking cities in Japan through Bayesian network analysis. The statistical urban model of this study indicates the cascading and complex relationships between the factors affecting population decline in shrinking cities. While the results regarding social indicators are in line with those of previous research [26–28], the current findings are still novel and unexpected; particularly, the study shows that educational indicators have a more substantial impact than economic indicators such as the financial strength index. Previous studies have pointed to the impact of economic decline on shrinking cities [22, 24]; this occurs when people migrate out of cities as local industry declines and jobs are lost. Meanwhile, the results of this study demonstrate that population decline in Japanese shrinking cities is not because of such outflow decline related to migration but because of a natural decline related to low birthrates and aging populations.

This reality of shrinking cities in Japan might explain why children's education have a stronger impact on the PCR in this context. Parents might feel more comfortable having and raising children if city governments value educational support. The educational support includes not only teachers but also educational computers. This result is consistent with those of prior research showing that those who continue to live in shrinking cities have been found to value the attractions of the city, including social connections, and economic activities in the city [37].

Still, this relevance of educational indicators might be unique to Japan, which has experienced a declining population due to its low birthrates. In Japan, education accounts for a low percentage of national and administrative fiscal expenditures, primarily because social security expenditures for older adults are a heavy burden [38]. However, when considering the current results and the fiscal expenditure limitations in Japan, the suggestion is that increasing investment in education for children might help solve the problem of shrinking cities.

The results also suggest that urban planning indicators—including "number of community centers," "number of medical clinics," "number of homes located more than 500 meters from the nearest elementary school," and "number of land transactions"—do not directly affect population decline rates in Japanese shrinking cities. Further, based on the low STE scores for these indicators, they only indirectly affect the population decline rate. This result is surprising when seen against studies that analyzed shrinking cities in the context of urban planning [28, 29]. However, these findings do not imply that urban planning has no impact on the population decline rate. Urban planners are still suggested to continue to implement urban planning strategies for shrinking cities. At the same time, considering that governments have limited financial resources, the results also suggest that it might not be necessary for governments to prioritize urban planning as a target for tackling population decline in shrinking cities. Instead, it might be better to effectively maintain and manage, not build and construct, the range of urban planning-related factors that were developed during the city's population growth period. Overall, urban planning in shrinking cities remains an important topic that needs to be considered more in the future.

It should be noted that this study was conducted in Japan, a country characterized by a comfort climate. However, in some countries worldwide, including those in the Global South, global warming issues might cause population decline. Particularly, researchers show that

there are correlations between the urban population, the built environment, and land surface temperatures [39–42], and that land surface temperatures are influenced more by the built than natural environment [43]. Another study shows that the relationship between these factors and urban temperature is nonlinear [44]. These pieces of evidence suggest that populations might begin to decline as temperatures exceed a threshold. Specifically, the natural disasters caused by global warming, such as hurricanes and floods, have been reported to be related to emerging refugee issues [45]. Thus, future researchers attempting to use methods and conduct analyses similar to those of the current study in the context of other countries may need to also consider the impact of global warming. Scholars have also remarked that research networks enabling the interaction between research and practice play an essential role in global research regarding shrinking cities [46].

A limitation of this study is the use of data only as of 2011. Specifically, the dataset of economic, social, and educational indicators used in this study covers most city-level government statistics; there is no other Japanese dataset that covers as many city-level indicators. However, due to the need to match the year, the data is only as of 2011. Because population decline is a long-term urban phenomenon, this dataset still allowed for the provision of novel conclusions. Considering this limitation, researchers could endeavor to combine the most recent and past datasets comprising economic, social, and educational indicators of cities into time series data, and then analyze the urban model developed in this study using this time series dataset. Such analysis may help us investigate future scenarios based on the urban model developed in this study.

## 5. Conclusions

In conclusion, the study findings provide essential insights into the importance of education investment for children in shrinking cities. Regarding theoretical implications, this study shows that the impact of educational investment is more substantial than that of economic indicators (e.g., the financial strength index). With this in mind and considering the limitations regarding fiscal expenditures in Japan, it may be important for municipal leaders to invest in education for children in order to prevent rapid population decline. In summary, increasing the budget for education to tackle the decline in the number of children and increasing the welfare expenditure to tackle the growth in the number of older adults are likely important measures in the context of population decline because of low birth rates and aging populations.

In Japan, population decline is expected to continue for a long time, as immigration is low. In this context, Japanese shrinking cities need to achieve a gradual, rather than a rapid, population decline to maintain residents' abundant lifestyle. The conclusions of this study could contribute to countries other than Japan that are likely to experience future population decline because of low birth rates and aging populations.

## Supporting information

**S1 Table. Results of the economic, social, and educational indicators.** The table below shows the 259 indicators with their average, standard deviation score, standardized total effect, mutual information, and p-value.
(PDF)

## Author Contributions

**Conceptualization:** Haruka Kato.

**Data curation:** Haruka Kato.

**Formal analysis:** Haruka Kato.

**Funding acquisition:** Haruka Kato.

**Investigation:** Haruka Kato.

**Methodology:** Haruka Kato.

**Project administration:** Haruka Kato.

**Resources:** Haruka Kato.

**Software:** Haruka Kato.

**Supervision:** Haruka Kato.

**Validation:** Haruka Kato.

**Visualization:** Haruka Kato.

**Writing – original draft:** Haruka Kato.

**Writing – review & editing:** Haruka Kato.

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
