## [Decision Letter · Decision Letter 0]

19 Jan 2023

PONE-D-22-35408Urban Modeling of Shrinking Cities through Bayesian Network Analysis Using Economic, Social, and Educational Indicators: Case of Japanese CitiesPLOS ONE

Dear Dr. Kato,

Thank you for submitting your manuscript to PLOS ONE. After careful consideration, we feel that it has merit but does not fully meet PLOS ONE’s publication criteria as it currently stands. Therefore, we invite you to submit a revised version of the manuscript that addresses the points raised during the review process.

We look forward to receiving your revised manuscript.

Kind regards,

Jun Yang

Academic Editor

PLOS ONE

Additional Editor Comments:

Reviewer 1

The authors analyzed the impacts of economic, social, and educational factors on population decline in Japanese shrinking cities. The disadvantages of this research are that the research content is simple, the structure is irrational, and the innovation is insufficient. I focus here only on some points, which are hopefully easy for the authors to take into account in the revision.

(1) Line 33-34, Fig.1 is not clear enough, and i do not understand population change rate in 2010. What does it mean?

(2) Line 37, 'However, these risks have not been identified systematically.', add more details.

(3) Sec Literature Review, conlude it and highlight the innovation.

(4) Sec Economic, Social, and Educational Indicators, why choose this these indicators, explain it. Importantly, Is there collinearity?

(5) Sec Discussions, this part should be improved. Does urban climate change affect city shrinkage and population. Some important references as follows:

1）Understanding seasonal contributions of urban morphology to thermal environment based on boosted regression tree approach, Building and Environment, 2022, 2022, 226: 109770, doi: 10.1016/j.buildenv.2022.109770.

2）Contribution of urban functional zones to the spatial distribution of urban thermal environment, Building and Environment (2022), doi: 10.1016/j.buildenv.2022.109000.

3）Modelling spatial distribution of fine-scale populations based on residential properties, International Journal of Remote Sensing (2019), doi: https://doi.org/10.1080/01431161.2019.1579387.

4）Exploring thermal comfort of urban buildings based on local climate zones, Journal of Cleaner Production (2022), doi: https://doi.org/10.1016/j.jclepro.2022.130744.

5）The impact of urban renewal on land surface temperature changes: A case study in the main city of Guangzhou, China. Remote Sensing (2020), https://doi.org/10.3390/rs12050794.

(6) Sec Conclusions, conclude the main colusions of this study, remove references.

(7) English language should be improved.

Reviewer 2

The author used bayesian network analysis to explored the influencing factors of shrinking cities from the perspective of economic, social and education. The topic is innovative, while there also have some problems should be revised as follow.

1.There are lack of the theory or logic framework to illustrate the relationship of shrinking cities with the indicators. The author should not only calculated the correlation of them.

2. There should have some formulas or models in methods.

3.The websites of data source should not only be added in references, but also needed to be added in data source.

4.In line 112, the authors mentioned population change rate. Is it refer to average population change rate in ten years or total population change rate in ten years.

5.In line 125, the authors illustrated there have 208 out of 259 indicators formed a significant network. However, there only have 14 indicators with STE more than 0.2. The author should listed all of correlation of 208 indicators.

6.What's the difference of section bayesian network of shrinking cities with correlation of indicatiors. Maybe these two sections could be integrated into one section.

7.The significance of correlation should be mentioned. Such as 99%,95% and so on.

8.In line 51, the authors could re-draw a new figure of study area to show the 1316 cities clearly by GIS. The color of study area and non-study area are different.

9.In table 1, what's the BMI refer to in different ranges? It should be explained clearly.

10.In conclusion, the author should illustrate some conclusions in your own research. What can be concluded by your research? It is not suitable to cite others researches.

11.The serial number of sub-title should be added.

Reviewers' comments:

Reviewer's Responses to Questions

**Comments to the Author**

1. Is the manuscript technically sound, and do the data support the conclusions?

Reviewer #1: Partly

Reviewer #2: Yes

2. Has the statistical analysis been performed appropriately and rigorously? 

Reviewer #1: Yes

Reviewer #2: Yes

3. Have the authors made all data underlying the findings in their manuscript fully available?

Reviewer #1: Yes

Reviewer #2: Yes

4. Is the manuscript presented in an intelligible fashion and written in standard English?

Reviewer #1: No

Reviewer #2: Yes

5. Review Comments to the Author

Reviewer #1: The authors analyzed the impacts of economic, social, and educational factors on population decline in Japanese shrinking cities. The disadvantages of this research are that the research content is simple, the structure is irrational, and the innovation is insufficient. I focus here only on some points, which are hopefully easy for the authors to take into account in the revision.

(1) Line 33-34, Fig.1 is not clear enough, and i do not understand population change rate in 2010. What does it mean?

(2) Line 37, 'However, these risks have not been identified systematically.', add more details.

(3) Sec Literature Review, conlude it and highlight the innovation.

(4) Sec Economic, Social, and Educational Indicators, why choose this these indicators, explain it. Importantly, Is there collinearity?

(5) Sec Discussions, this part should be improved. Does urban climate change affect city shrinkage and population. Some important references as follows:

1）Understanding seasonal contributions of urban morphology to thermal environment based on boosted regression tree approach, Building and Environment, 2022, 2022, 226: 109770, doi: 10.1016/j.buildenv.2022.109770.

2）Contribution of urban functional zones to the spatial distribution of urban thermal environment, Building and Environment (2022), doi: 10.1016/j.buildenv.2022.109000.

3）Modelling spatial distribution of fine-scale populations based on residential properties, International Journal of Remote Sensing (2019), doi: https://doi.org/10.1080/01431161.2019.1579387.

4）Exploring thermal comfort of urban buildings based on local climate zones, Journal of Cleaner Production (2022), doi: https://doi.org/10.1016/j.jclepro.2022.130744.

5）The impact of urban renewal on land surface temperature changes: A case study in the main city of Guangzhou, China. Remote Sensing (2020), https://doi.org/10.3390/rs12050794.

(6) Sec Conclusions, conclude the main colusions of this study, remove references.

(7) English language should be improved.

Reviewer #2: The author used bayesian network analysis to explored the influencing factors of shrinking cities from the perspective of economic, social and education. The topic is innovative, while there also have some problems should be revised as follow.

1.There are lack of the theory or logic framework to illustrate the relationship of shrinking cities with the indicators. The author should not only calculated the correlation of them.

2. There should have some formulas or models in methods.

3.The websites of data source should not only be added in references, but also needed to be added in data source.

4.In line 112, the authors mentioned population change rate. Is it refer to average population change rate in ten years or total population change rate in ten years.

5.In line 125, the authors illustrated there have 208 out of 259 indicators formed a significant network. However, there only have 14 indicators with STE more than 0.2. The author should listed all of correlation of 208 indicators.

6.What's the difference of section bayesian network of shrinking cities with correlation of indicatiors. Maybe these two sections could be integrated into one section.

7.The significance of correlation should be mentioned. Such as 99%,95% and so on.

8.In line 51, the authors could re-draw a new figure of study area to show the 1316 cities clearly by GIS. The color of study area and non-study area are different.

9.In table 1, what's the BMI refer to in different ranges? It should be explained clearly.

10.In conclusion, the author should illustrate some conclusions in your own research. What can be concluded by your research? It is not suitable to cite others researches.

11.The serial number of sub-title should be added.

6. PLOS authors have the option to publish the peer review history of their article (what does this mean?). If published, this will include your full peer review and any attached files.

Reviewer #1: No

Reviewer #2: No

---

## [Author Response · Author response to Decision Letter 0]

8 Mar 2023

Dear Reviewer:

We appreciate the reviewer for the generous comment on the manuscript. We have attached our response letter. We believe that the manuscript is now suitable for publication in Plos One and look forward to hearing from you concerning your decision.

Yours sincerely

---

## [Decision Letter · Decision Letter 1]

15 Mar 2023

PONE-D-22-35408R1Urban Modeling of Shrinking Cities through Bayesian Network Analysis Using Economic, Social, and Educational Indicators: Case of Japanese CitiesPLOS ONE

Dear Dr. Kato,

Thank you for submitting your manuscript to PLOS ONE. After careful consideration, we feel that it has merit but does not fully meet PLOS ONE’s publication criteria as it currently stands. Therefore, we invite you to submit a revised version of the manuscript that addresses the points raised during the review process.

We look forward to receiving your revised manuscript.

Kind regards,

Jun Yang

Academic Editor

PLOS ONE

Journal Requirements:

Additional Editor Comments:

The quality of this manuscript have improved after revision. There only have two problems should be revised. 1.The authors choose population change rate aged 0-14 and population aged over 65. Why the authors not choose age 15-64. It should be explained simply. 2.Some relevant references should be cited. Spatial evolution of population change in Northeast China during 1992–2018. Science of the Total Environment.2021,776:146023.https://doi.org/10.1016/j.scitotenv.2021.146023. Spatial and temporal heterogeneity of urban land area and PM2.5 concentration in China.Urban Climate,2022,45:101268.doi:https://doi.org/10.1016/j.uclim.2022.101268. Spatial-Temporal Characteristics of Primary and Secondary Educational Resources for Relocated Children of Migrant Workers:The Case of Liaoning Province.Complexity,volume 2020,Article ID7457109,13 pages.doi:https://doi.org/10.1155/2020/7457109. Theoretical framework and research prospect of the impact of China's digital economic development on population.Frontiers in Earth Science,2022,10:988608. doi: 10.3389/feart.2022.988608. Differences in Accessibility of Public Health Facilities in Hierarchical Municipalities and the Spatial Pattern Characteristics of Their Services in Doumen District,China.Land 2021, 10, 1249. https://doi.org/10.3390/land10111249. Spatiotemporal relationship characteristic of climate comfort of urban human settlement environment and population density in China. Front. Ecol. Evol. 2022,10:953725.doi: 10.3389/fevo.2022.953725. Spatio–temporal evolution and factors of climate comfort for urban human settlements in the Guangdong–Hong Kong–Macau Greater Bay Area. Front. Environ. Sci. 2022,10:1001064. doi: 10.3389/fenvs.2022.1001064

Reviewers' comments:

Reviewer's Responses to Questions

**Comments to the Author**

1. If the authors have adequately addressed your comments raised in a previous round of review and you feel that this manuscript is now acceptable for publication, you may indicate that here to bypass the “Comments to the Author” section, enter your conflict of interest statement in the “Confidential to Editor” section, and submit your "Accept" recommendation.

Reviewer #1: (No Response)

Reviewer #2: All comments have been addressed

2. Is the manuscript technically sound, and do the data support the conclusions?

Reviewer #1: (No Response)

Reviewer #2: Yes

3. Has the statistical analysis been performed appropriately and rigorously? 

Reviewer #1: (No Response)

Reviewer #2: Yes

4. Have the authors made all data underlying the findings in their manuscript fully available?

Reviewer #1: (No Response)

Reviewer #2: Yes

5. Is the manuscript presented in an intelligible fashion and written in standard English?

Reviewer #1: (No Response)

Reviewer #2: Yes

6. Review Comments to the Author

Reviewer #1: (No Response)

Reviewer #2: The quality of this manuscript have improved after revision. There only have two problems should be revised.

1.The authors choose population change rate aged 0-14 and population aged over 65. Why the authors not choose age 15-64. It should be explained simply.

2.Some relevant references should be cited.

Spatial evolution of population change in Northeast China during 1992–2018. Science of the Total Environment.2021,776:146023.https://doi.org/10.1016/j.scitotenv.2021.146023.

Spatial and temporal heterogeneity of urban land area and PM2.5 concentration in China.Urban Climate,2022,45:101268.doi:https://doi.org/10.1016/j.uclim.2022.101268.

Spatial-Temporal Characteristics of Primary and Secondary Educational Resources for Relocated Children of Migrant Workers:The Case of Liaoning Province.Complexity,volume 2020,Article ID7457109,13 pages.doi:https://doi.org/10.1155/2020/7457109.

Theoretical framework and research prospect of the impact of China's digital economic development on population.Frontiers in Earth Science,2022,10:988608. doi: 10.3389/feart.2022.988608.

Differences in Accessibility of Public Health Facilities in Hierarchical Municipalities and the Spatial Pattern Characteristics of Their Services in Doumen District,China.Land 2021, 10, 1249. https://doi.org/10.3390/land10111249.

Spatiotemporal relationship characteristic of climate comfort of urban human settlement environment and population density in China. Front. Ecol. Evol. 2022,10:953725.doi: 10.3389/fevo.2022.953725.

Spatio–temporal evolution and factors of climate comfort for urban human settlements in the Guangdong–Hong Kong–Macau Greater Bay Area. Front. Environ. Sci. 2022,10:1001064. doi: 10.3389/fenvs.2022.1001064

7. PLOS authors have the option to publish the peer review history of their article (what does this mean?). If published, this will include your full peer review and any attached files.

Reviewer #1: No

Reviewer #2: No

---

## [Author Response · Author response to Decision Letter 1]

23 Mar 2023

Thank you for giving us the opportunity to strengthen our manuscript with your valuable comments. I have worked hard to incorporate your feedback and hope that these revisions persuade you to accept our submission.

---

## [Decision Letter · Decision Letter 2]

27 Mar 2023

Urban Modeling of Shrinking Cities through Bayesian Network Analysis Using Economic, Social, and Educational Indicators: Case of Japanese Cities

PONE-D-22-35408R2

Dear Dr. Kato,

We’re pleased to inform you that your manuscript has been judged scientifically suitable for publication and will be formally accepted for publication once it meets all outstanding technical requirements.

Kind regards,

Jun Yang

Academic Editor

PLOS ONE

Additional Editor Comments (optional):

Accept

Reviewers' comments:

Reviewer's Responses to Questions

**Comments to the Author**

1. If the authors have adequately addressed your comments raised in a previous round of review and you feel that this manuscript is now acceptable for publication, you may indicate that here to bypass the “Comments to the Author” section, enter your conflict of interest statement in the “Confidential to Editor” section, and submit your "Accept" recommendation.

Reviewer #1: (No Response)

Reviewer #2: All comments have been addressed

2. Is the manuscript technically sound, and do the data support the conclusions?

Reviewer #1: (No Response)

Reviewer #2: Yes

3. Has the statistical analysis been performed appropriately and rigorously? 

Reviewer #1: (No Response)

Reviewer #2: Yes

4. Have the authors made all data underlying the findings in their manuscript fully available?

Reviewer #1: (No Response)

Reviewer #2: Yes

5. Is the manuscript presented in an intelligible fashion and written in standard English?

Reviewer #1: (No Response)

Reviewer #2: Yes

6. Review Comments to the Author

Reviewer #1: (No Response)

Reviewer #2: The authors have revised all the problems. All the problems have been addressed. I think it could be accepted.

7. PLOS authors have the option to publish the peer review history of their article (what does this mean?). If published, this will include your full peer review and any attached files.

Reviewer #1: No

Reviewer #2: No

---

## [Editor Report · Acceptance letter]

31 Mar 2023

PONE-D-22-35408R2 

Urban Modeling of Shrinking Cities through Bayesian Network Analysis Using Economic, Social, and Educational Indicators: Case of Japanese Cities 

Dear Dr. Kato:

I'm pleased to inform you that your manuscript has been deemed suitable for publication in PLOS ONE. Congratulations! Your manuscript is now with our production department. 

Kind regards, 

on behalf of

Dr. Jun Yang 

Academic Editor

PLOS ONE